# Comments by Microbiologists for Interpreting Antimicrobial Susceptibility Testing and Improving the Appropriateness of Antibiotic Therapy in Community-Acquired Urinary Tract Infections: A Randomized Double-Blind Digital Case-Vignette Controlled Superiority Trial

**DOI:** 10.3390/antibiotics12081272

**Published:** 2023-08-02

**Authors:** Emilie Piet, Youssoupha N’Diaye, Johann Marzani, Lucas Pires, Hélène Petitprez, Tristan Delory

**Affiliations:** 1Department of Infectious Diseases, Centre Hospitalier Annecy Genevois, 74370 Epagny Metz-Tessy, France; 2Clinical Research Department, Centre Hospitalier Annecy Genevois, 74370 Epagny Metz-Tessy, France; 3Microbiological Analysis Department, Centre Hospitalier Annecy Genevois, 74370 Epagny Metz-Tessy, France

**Keywords:** urinary tract infection, urinalysis, antibiotic prescribing, primary care, antimicrobial stewardship, appropriateness, digital trial

## Abstract

In primary care, urinary tract infections (UTIs) account for the majority of antibiotic prescriptions. Comments from microbiologists on interpreting the antimicrobial susceptibility testing (AST) profile for urinalysis were made to improve the prescription of antibiotics. We aimed to explore the added value of these comments on the quality of antibiotic prescribing by a superior double-blind digital randomized case-vignette trial among French general practitioners (GPs). One case vignette with (intervention) or without (control) a ‘comment’ after AST was randomly assigned to GPs. Among 815 participating GPs, 64.7% were women, at an average age of 37 years. Most (90.1%) used a computerized decision support system for prescribing antibiotics. Empirical antibiotic therapy was appropriate in 71.9% (95% CI, 68.8–75.0) of the cases, without differences between arms. The overall appropriateness of targeted antibiotic therapy (primary outcome) was not significantly increased when providing ‘comments’: 83.4% vs. 79.9% (OR = 1.26, 95% CI, 0.86–1.85). With the multivariate analysis, the appropriateness was improved by 2-folds (OR = 2.38, 95% CI, 1.02–6.16) among physicians working in healthcare facilities. Among digital-affine young general practitioners, the adjunction of a ‘comment’ by a microbiologist to interpret urinalysis in community-acquired UTIs did not improve the overall level of appropriateness of the targeted antibiotic.

## 1. Introduction

The burden of antimicrobial resistance has dramatically increased over the last few years worldwide [1,2,3]. In a One Health approach, Antibiotic Stewardship Programs are implemented throughout national action plans to ensure the rational use of antibiotics and limit the spread of multiresistant bacteria. These programs involve domains such as medicinal management and prescribing systems, or technology to optimize antibiotic prescribing and use [4]. In high-income countries, 70–90% of antibiotic therapies are initiated by general practitioners (GPs), resulting in high antibiotic consumption in primary care [5,6]. In France in 2021, the consumption of antibiotics in the community was as high as 20 defined daily doses per 1000 inhabitants per day, while the average European consumption is 15 [7].

GPs are therefore a key target for Antibiotic Stewardship Programs. However, hospital-based interventions that have been developed over the past decade are not directly scalable to the primary care setting, and innovative interventions tailored to these settings are needed. To that end, computerized decision support systems (CDSSs) [8], or other interventions for optimized antibiotic prescribing, may be useful. In general, CDSSs are used to support clinicians in their complex decision-making processes by linking clinical observation and patient information to available targeted and specific clinical knowledge [9]. CDSSs for antibiotic prescribing are often active interventions that give specific recommendations, and few are restricting antibiotics [10].

In primary care, urinary tract infections (UTIs) account for most antibiotic prescriptions [5,11,12,13,14]. In the early 2010s, up to 80% of empiric antibiotic therapies initiated for UTIs were deemed unnecessary or inappropriate [6,15,16,17]. The use of broad-spectrum oral antibiotics such as fluoroquinolones was frequent, ranging from 44 to 60% of the cases, resulting in up to 77% of inappropriate antibiotic therapy [17]. However, a recent study in a family practice in Switzerland showed increased adherence to guidelines in UTIs, with only 14% inappropriate antibiotic therapy [18]. A recent case-vignette trial showed that CDSSs providing recommendation for diagnosis and therapy for pyelonephritis improved GPs’ adherence to guidelines [16]. In the Netherlands, the use of a CDSSs to estimate the probability of success of eight commonly used antibiotics for treating UTIs from electronic health record data, based on machine-learning algorithms, decreased the rate of antibiotic re-prescription 28 days after initial treatment [19]. In France, the ‘Antibioclic’ CDSS is being widely used by GPs [20,21]. However, evidence of its impact on antimicrobial consumption and resistance is lacking.

The urinalysis is the key diagnostic test for identifying the bacterium/bacteria involved in a UTI and to establish its/their antimicrobial susceptibility testing (AST) profile. A urinalysis allows GPs to select a targeted and effective documented antibiotic therapy, ideally with the narrowest spectrum possible, to reduce the risk of emerging resistance [22]. However, in primary care in France, most antibiotic therapies are initiated empirically, without urinalysis results, and less than a third of documented UTIs result in appropriate antibiotic therapies [17,23]. It has been recommended in the United States, Australia, and France that comments by microbiologists for interpreting AST could be used to guide and improve the prescription of antibiotics; however, no clinical evidence is available [24,25,26].

We aimed at exploring the impact of comments for interpreting urinalysis AST, issued by primary care microbiologists, on the validity of prescribed antibiotics for community-acquired UTIs managed by GPs, assuming that it would guide GPs toward better antibiotic prescribing. Herein, we assess the comparative effect of such a comments-based versus standard urinalysis/AST report in regard to the appropriateness of targeted antibiotic therapy initiated by GPs, adjusting for confounding factors, including the use of a CDSS.

## 2. Results

Ten out of one hundred general medical councils accepted our request to broadcast our trial to their mailing list between 24 November 2020 and 6 June 2021. We enrolled 1015 GPs in the trial, mostly from the Auvergne-Rhône-Alpes area (52.3%), of whom 815 completed an interpretable questionnaire for empirical antibiotic therapy (intention-to-treat set) and 716 for targeted antibiotic therapy (by-the-protocol set). The GPs’ mean age was 37 years, and a majority (64.7%) of them were women. Most (73.8%) of the GPs were either established or replacing GPs, with an average professional experience of 11 years. The demographic and professional characteristics of GPs enrolled in the trial were similar between arms and are described in Table 1.

### 2.1. Attitudes toward Antimicrobial Susceptibility Testing

Table 2 displays the GPs’ attitudes toward the interpretation of a urinalysis and their use of the ‘Antibioclic’ CDSS or need for assistance from an infectious disease physician to manage community-acquired UTIs. No differences were observed between arms. Most of the GPs were comfortable with interpreting the AST (96.9%), and only a third were willing to receive specific training about AST interpretation. Less than a tenth of GPs were not using the ‘Antibioclic’ CDSS to prescribe antibiotics in community-acquired UTIs. GPs not using the CDSS were older (54 years of age, on average) than others (35 years of age, on average). Only a fifth sought the advice of an infectious disease specialist to manage UTIs.

### 2.2. Empirical Antibiotic Therapy

The appropriateness of empirical antibiotic therapy on the whole (intention to treat) trial’s population with interpretable questionnaire was 71.9% (95% CI, 68.8% to 75.0%), without differences between arms.

### 2.3. Targeted Antibiotic Therapy, Primary Outcome

We assessed the appropriateness of documented antibiotic therapy on the by-protocol population. This subset population included all GPs with an interpretable questionnaire for the primary outcome, excluding 99 (12.1%) GPs with missing information about the targeted antibiotic therapy. A total of 716 GPs were included in the primary outcome analysis: 343 in the intervention arm and 373 in the control arm. The overall appropriateness of targeted antibiotic therapy was not significantly increased when providing ‘comments’ for urinalysis interpretation: 83.4% vs. 79.9% (OR = 1.26, 95% CI, 0.86 to 1.85, *p*-value = 0.230).

Figure 1 shows the appropriateness of targeted antibiotic therapy by trial arm and type of UTI. It was the highest in asymptomatic bacteriuria in pregnant women (99.0%), in simple pyelonephritis (90.6%) or in pyelonephritis at risk of complication (88.0%), and in male UTIs (94.2%). The appropriateness was the lowest in cystitis (70.1%) and in bacteriuria associated with an indwelling urinary catheter (52.9%). The intervention did not increase appropriateness in any of these cases (*p*-value > 0.050).

In the multivariable sensitivity analysis conducted on 710 GPs with complete data (complete cases), the effect of intervention was similar for the two arms, when adjusting for GPs’ sex, age group, main mode of practice, professional experience, student mentoring status, CDSS use on appropriateness, and the type of UTI. The imputation of missing data by the MICE (multiple imputation by chained equation) method generated a larger number of inappropriate prescriptions and tended toward a more even distribution between the two randomization arms, without difference in point estimates.

GPs working in hospital settings had higher appropriateness than those working in primary care (OR = 2.38, 95% CI, 1.02 to 6.16), Table 3.

### 2.4. Deviation from First-Line Antibiotic Regimen, Secondary Outcome

The use of ‘comments’ did not reduce the deviation from the antibiotic regimen recommended in the first line, yielding a 0.07 non-significant absolute reduction and corresponding to a relative reduction by 6% in favor of the intervention arm (OR = 0.94, 95% CI, 0.86 to 1.01, *p*-value = 0.108).

### 2.5. Use of Broad-Spectrum Antibiotics, Secondary Outcome

In addition, adding a ‘comment’ did not reduce the prescribing rate of broad-spectrum antibiotics (OR = 0.92, 95% CI, 0.68 to 1.24, *p*-value = 0.587), especially those from the WATCH list [27]: 22.4% vs. 19.0% in control for amoxicillin–clavulanic acid, 6.3% vs. 6.8% in control for third-generation cephalosporins, and 71.3% vs. 73.6% in control for fluoroquinolones.

## 3. Discussion

Among young general practitioners who extensively use a CDSS to prescribe antibiotics in community-acquired UTIs, the delivery of a ‘comment’ by microbiologists to better interpret the urinalysis report did not improve the overall level of appropriateness of targeted antibiotic therapy.

This is the first study that attempted to explore the effect of such ‘comments’ on the appropriateness of antibiotic prescribing for documented urinary tract infections, while the interest for microbiologists-based recommendations has existed for 40 years [24].

In our digital trial, the appropriateness of empirical antibiotic therapy was very high (72%) compared to the 20–60% reported in the literature [16,17,23,28]. It even reached 80% for targeted antibiotic therapies in the control arm, suggesting that the GPs participating in the trial were not representative of the GP population as a whole. Digital trials do reach a digital affine population and may lack representativeness. To prevent this sampling bias, we tried to reach out to every GP registered to the ten departmental medical order boards that accepted to participate in the study. However, our population was younger (37 vs. 51 years of age), with higher representativeness of females (65% versus 59%) than that recorded for French GPs [29,30]. In fact, young physicians have been shown to have a higher rate of appropriate antibiotic prescribing [31]. Most of the participating GPs (90%) were using a web-based CDSS for antibiotic prescribing, while it is estimated that ~60% of French GPs are using it in real life (unpublished data). As in previous digital trials on selective AST reporting, the GPs were unequivocally aware of the clinical diagnosis, though we ensured the concealment of intervention [23,32]. For CDSS users, it was thereby possible to use the CDSS alongside the trial’s participation, and most of the CDSS users (93%) report following the CDSS’s recommendations when using it [20]. Moreover, since 2017, the ‘Antibioclic’ CDSS has included a module allowing GPs to use an AST report to target the selection of antibiotics [33]. This may have contaminated the trial and resulted in an overestimation of appropriateness in both arms. We may also have overestimated the appropriateness levels because we did not focus on the selected dose and duration of antibiotic therapies [6,16]. There was, however, a low likelihood of demonstrating a clinically relevant increase in appropriateness because of intervention, and the study was ultimately unpowered. At least 3900 GPs would have been needed to show a 5% absolute increase in appropriateness, from 80% to 85%.

During the last decade, many studies showed that selective AST reporting may improve the appropriateness of documented antibiotic therapy in community-acquired UTIs [32,34,35,36,37]. In the digital trial conducted by Coupat et al., a decrease in the intended prescription for fluoroquinolones in UTIs was observed when using selective AST reporting [32]. We did not observe any decrease in the intended prescription of broad-spectrum antibiotics by GPs receiving the intervention. Asymptomatic bacteriuria associated with an indwelling urinary catheter was the situation with the least appropriateness for antibiotic therapy. It shows the tendency of GPs to misuse antibiotics in this indication [22,38].

Additional limitations were due to software for setting the randomization, as it was not possible to stratify randomization regarding CDSS use. However, the concealment of allocation was ensured, and the randomization process allowed us to properly balance case vignettes between arms. Moreover, we did not use a user-centered approach in our pilot study conducted before the trial to ensure that the understanding of ‘comments’ was optimal [39]. Finally, although we did not conduct the trial in real-life settings, digital trials are deemed acceptable for the evaluation of interventions on antibiotic appropriateness [40,41].

## 4. Materials and Methods

### 4.1. Design, Participants, and Data Collection

We conducted a randomized (ratio 1:1) double-blind digital case-vignette controlled superiority trial among French GPs between 24 November 2020 and 6 June 2021.

Any French GPs, including residents, were eligible for single participation in the trial. Other medical specialists and medical students were excluded.

We sent an advertising email presenting the trial to the 100 French general medical councils, 30 regional residents’ unions, and 13 regional seniors’ physician unions. We invited organizations to broadcast the offer for participation in the trial to their mailing lists, using a preformatted email. The email included a web-link redirecting to a questionnaire hosted on the LimeSurvey web platform. The web link was active over the whole study period, and a dunning email was sent out 3 weeks after the initial broadcasting.

Upon reaching the trial’s online platform, the respondents answered a captcha (Turing test) to ensure that they were not bots. They were then screened for eligibility for the trial. We attached a unique anonymized tracking number to each participant and internet protocol address to ensure data confidentiality and prevent duplicates. Participants underwent two-step randomization that was centralized and operated by the trial platform. Allocation was concealed. The first step allocated participants to one either the comments (intervention) arm or the control arm. The second step allocated participants within arms to one of the six case vignettes designed for the trial. GPs were aware of the clinical diagnosis associated with the case vignette. The case vignettes are summarized in Appendix A. Case-1 was a complicated cystitis, Case-2 was a male urinary tract infection, Case-3 was an uncomplicated pyelonephritis, Case-4 was pyelonephritis at risk of complications, Case-5 was an asymptomatic bacteriuria in a pregnant woman, and Case-6 was an asymptomatic bacteriuria on indwelling urinary catheter. To ensure that the randomization procedure would be balanced between arms, we simulated 500 cases in the trial pilot phase. A minimum of 150 cases were needed to balance randomization between arms, and 300 for balance within arms.

### 4.2. Self-Questionnaire

After randomization, we asked GPs to fill out a web self-questionnaire divided into three parts. The first part (Part I) was common to any participant in the trial. In Part I, the GPs had to report information about their demographics, their experience in using and interpreting AST, and their use of the ‘Antibioclic’ web-based CDSS for prescribing antibiotics in primary care [20]. The second (Part II) and the third (Part III) parts differed according to the two-step randomization procedure. In Part II, GPs had to report their willingness to initiate empirical antibiotic therapy, and those willing to had to select the antibiotic by selecting the international non-proprietary name (INN) from a pulldown menu listing. Dose and duration were not reported. Those not willing to prescribe empirical antibiotic therapy were shifted to Part III. In Part III, we displayed the results of AST by allocated case vignette to the GPs. To GPs randomized in the intervention arm, we displayed a comment for urinalysis interpretation issued by a microbiologist (Appendix A). To others (control arm), we showed the crude report of AST without comment for urinalysis interpretation. We asked all GPs to report their willingness to prescribe a targeted antibiotic therapy, and if they were willing to, we collected the selected antibiotic using the procedure described in Part II. We tested survey questionnaires in a pilot phase on 15 GPs for validation. Figure 2 shows the study flow diagram.

### 4.3. Comments, i.e., the Intervention

The ‘comments’ consisted of predefined texts to be displayed below the crude report results of the AST (Appendix A). The crude results of susceptibility testing were similar between arms but varied by case vignette. We selected ‘comments’ among a pool established in 2018 by a panel of six microbiologists and one infectious disease physician participating in the hospital–town network established by the Department of Infectious Diseases at the Annecy Genevois hospital. Each ‘comment’ provides a hierarchical classification of appropriate antibiotics by type of UTI, from first to last lines, according to 2018 French guidelines for the management of community-acquired UTIs [22].

### 4.4. Outcomes

The primary outcome was the overall appropriateness of the targeted antibiotic therapy prescribed according to previously established 2018 French guidelines for the management of community-acquired UTIs [22]. The targeted antibiotic therapy could be either appropriate (value 1) or inappropriate (value 0) according to the guidelines. We estimated the appropriateness of targeted antibiotic therapy by arm. The secondary outcomes were as follows: (i) The first secondary outcome was the degrees of deviation from the first-line antibiotic therapy that was recommended. When the prescribed antibiotic corresponded to the first-line treatment recommended, we assumed that there was no deviation (0 degrees of deviation). Otherwise, we added one degree of deviation per increase in line number, up to a maximum of 4 degrees of deviation between the first and fifth lines [22]. (ii) The second secondary outcome was the rate of broad-spectrum antibiotics prescription: amoxicillin–clavulanate, third-generation cephalosporins, or fluoroquinolones.

### 4.5. Statistical Analysis

Before the trial started, we estimated that a minimum of 500 answers would allow us to reach 90% power to detect an increase in overall appropriateness, from 65% in the control arm to 78% in the intervention arm.

We conducted the main analysis on the by-the-protocol population, enrolling all participants for which the primary outcome was available. We used binomial logistic regression to compare the levels of appropriateness of targeted antibiotic therapy between arms. We also conducted a set of sensitivity analyses by (i) conducting a multivariable binomial logistic regression assessing the overall appropriateness among GPs with complete data, by adjusting for potential confounders—demographic characteristics, use of CDSS, and assigned case vignette; and (ii) by imputing missing data for the primary outcome, using baseline characteristics and empirical antibiotic therapy in multiple imputation by chained equation (MICE). We used a log-transformation to report the difference between the general linear models as the odds ratio (OR) and its 95% confidence interval (95% CI). All tests were two-sided at a 5% threshold (*p*-value < 0.050) to indicate statistical significance. We performed the analysis using the R software version 4.03 (R foundation for Statistical Computing, Vienna, Austria).

## 5. Conclusions

The comments made by microbiologists regarding urinalysis and the interpretation of antimicrobial susceptibility testing in community-acquired UTIs did not improve the selection of an adequate targeted antibiotic therapy in a young and digital-affine population of GPs extensively using a web-based CDSS to prescribe antibiotics. Such approaches deserve randomized real-life investigations before being prioritized as a component of national antimicrobial action plans.

## Figures and Tables

**Figure 1 antibiotics-12-01272-f001:**
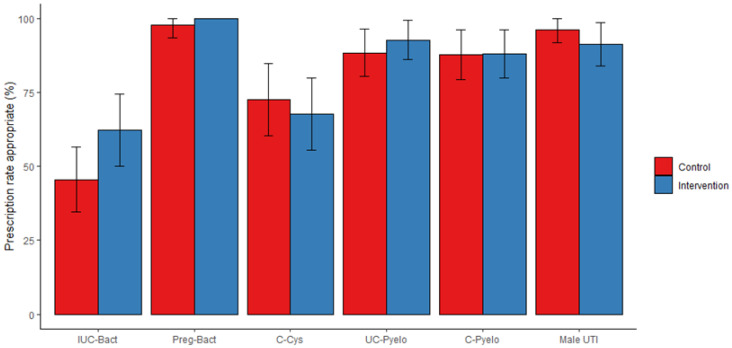
Appropriateness rate of documented antibiotic therapy according to type of urinary tract infection. Appropriateness rates are displayed using bars, with the 95% confidence interval at the top. The rates were computed for the by-protocol population (N = 716). Red bars are used for the control arm, and blue bars for the intervention arm. IUC-Bact: bacteriuria associated with an indwelling urinary catheter. Preg-Bact: bacteriuria in pregnancy. C-Cys: complicated cystitis. UC-Pyelo: uncomplicated pyelonephritis. C-Pyelo: pyelonephritis at risk of complication. Male-UTI: male urinary tract infection.

**Figure 2 antibiotics-12-01272-f002:**
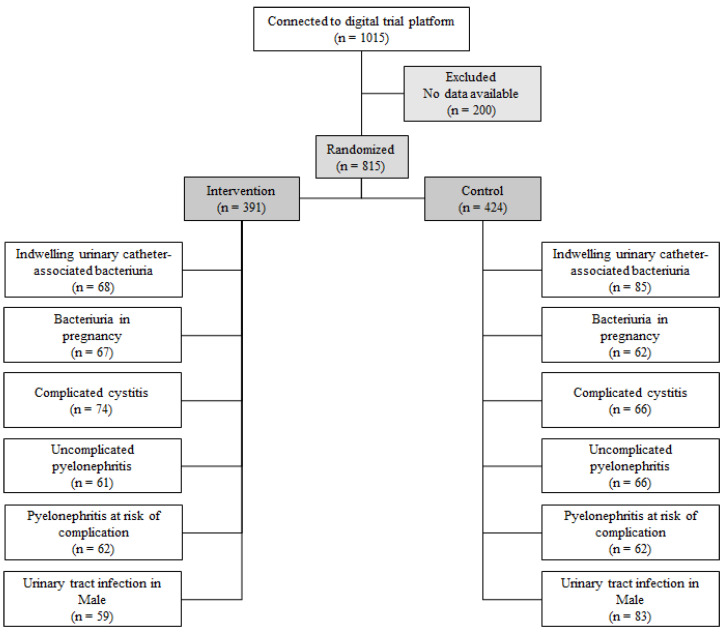
Flow diagram.

**Table 1 antibiotics-12-01272-t001:** Demographic characteristics of general practitioners, by trial arm.

	Control (N = 424)	Intervention (N = 391)	Total (N = 815)
**Sex**			*N* = 812
Women	280 (66.5%)	245 (62.7%)	525 (64.7%)
Men	141 (33.5%)	146 (37.3%)	287 (35.3%)
**Age**			*N* = 815
Mean ± SD	37.1 ± 11.6	36.7 ± 11.2	36.9 ± 11.4
Median (Q1–Q3)	33.0 (29.0–42.0)	33.0 (29.0–39.5)	33.0 (29.0–41.0)
Min–max	23.0–73.0	24.0–74.0	23.0–74.0
**Professional status of GPs ***			*N* = 815
Established	258 (60.8%)	237 (60.6%)	495 (60.7%)
Replacing other GPs	53 (12.5%)	54 (13.8%)	107 (13.1%)
Resident	113 (26.7%)	100 (25.6%)	213 (26.2%)
**Professional experience**			*N* = 815
Mean ± SD	11.7 ± 11.8	10.6 ± 11.3	11.2 ± 11.6
Median (Q1–Q3)	7.00 (3.00–19.0)	6.00 (3.00–16.0)	6.00 (3.00–17.0)
Min–max	0–48.0	0–46.0	0–48.0
**Working environment**			*N* = 814
Urban	156 (36.9%)	148 (37.9%)	304 (37.3%)
Semi-rural	208 (49.2%)	187 (47.8%)	395 (48.5%)
Rural	59 (13.9%)	56 (14.3%)	115 (14.1%)
**Main mode of practice**			*N* = 815
At hospital/healthcare facility	41 (9.67%)	32 (8.18%)	73 (8.96%)
In group practices	330 (77.8%)	318 (81.3%)	648 (79.5%)
Alone	53 (12.5%)	41 (10.5%)	94 (11.5%)
**Student mentoring**			*N* = 815
Yes	104 (24.5%)	82 (21.0%)	186 (22.8%)
No	320 (75.5%)	309 (79.0%)	629 (77.2%)

* General practitioners.

**Table 2 antibiotics-12-01272-t002:** Professional practices of general practitioners for management of urinary tract infections, by trial arm.

	Control (N = 424)	Intervention (N = 391)	Total (N = 815)
**Attitude toward AST * interpretation**			
At ease to interpret; willing to be trained	135 (31.8%)	112 (28.7%)	247 (30.3%)
At ease to interpret; not willing to be trained	274 (64.6%)	269 (68.9%)	543 (66.6%)
Not at ease to interpret; willing to be trained	12 (2.8%)	9 (2.3%)	21 (2.7%)
Not at ease to interpret; not willing to be trained	3 (0.8%)	1 (0.1%)	4 (0.4%)
**CDSS ^†^ use for prescribing antibiotics in UTIs ^‡^**			
Yes	383 (90.3%)	351 (89.8%)	734 (90.1%)
No	41 (9.67%)	40 (10.2%)	81 (9.94%)
**Seek advice from an infectious disease specialist for management of UTIs ^‡^**			
Frequently	5 (1.18%)	4 (1.02%)	9 (1.10%)
Occasionally	87 (20.5%)	69 (17.6%)	156 (19.1%)
Rarely	247 (58.3%)	225 (57.5%)	472 (57.9%)
Never	85 (20.0%)	93 (23.8%)	178 (21.8%)

*** Antimicrobial susceptibility testing. ^†^ Computerized decision support system. ^‡^ Urinary tract infections.

**Table 3 antibiotics-12-01272-t003:** General practitioners’ characteristics associated with the appropriateness of targeted antibiotic therapy and multivariable analysis.

Variable	N	OR *	95% CI	*p*-Value
**Sex**				
Women	452	Ref.		
Man	258	1.12	0.71–1.78	0.600
**Age (categories)**				
23 to 34 years-old	425	Ref.		
35 to 74 years-old	285	0.91	0.42–1.91	0.800
**Main mode of practice**				
In primary care	646	Ref.		
In healthcare facilities	64	2.38	1.02–6.16	0.046
**Professional experience**				
>5 years	319	Ref.		
≤5 years	200	0.71	0.32–1.53	0.400
Resident	191	0.82	0.34–1.92	0.600
**Student mentoring**				
Yes	167	Ref.		
No	543	1.08	0.63–1.84	0.800
**CDSS ^†^ use for antibiotic prescribing**				
Yes	643	Ref.		
No	37	0.76	0.36–1.69	0.500

* Odds ratio. The analysis is adjusted for randomization arm (intervention vs. control) and the allocated case-vignette. ^†^ Computerized decision support system, namely ‘Antibioclic’.

## Data Availability

According to the consent signed by participants, the data can be shared upon request to the corresponding author.

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
