# Peer review of "Comments by Microbiologists for Interpreting Antimicrobial Susceptibility Testing and Improving the Appropriateness of Antibiotic Therapy in Community-Acquired Urinary Tract Infections: A Randomized Double-Blind Digital Case-Vignette Controlled Superiority Trial"

_antibiotics, 2023, doi:10.3390/antibiotics12081272_

Round 1

Reviewer 1 Report

Dear Authors,

I do not have any critical remarks on the paper you submitted. The paper presents an interesting aspect of antibiotic therapy administration and the assessment of it.

The paper is written in a clear and understandable language, I found some minor issues (e.g. the first sentence of the Abstract needs correction, that's it). Even though this paper does not present any groundbreaking discoveries, it is an impoerant voice in assessing the causes of widely spreading antimicrobial resistance.

The language used in the paper is clear and easy to understand. I found one sentence that needs correction (first sentence of the Abstract) and that's all.

Author Response

Dear reviewer,

Thank you for your comment. We changed the first sentence of the abstract.

Reviewer 2 Report

Dear Colleagues

I reviewer your articles “Comments by microbiologists for interpreting antimicrobial susceptibility testing and improving the appropriateness of antibiotic therapy in community-acquired urinary tract infections: a randomized, double-blind, digital, case-vignette, controlled, …..”. Your results demonstrated that the primary care, urinary tract infections (UTIs) originate most of antibiotic prescribing by GPs mostly from the Auvergne-Rhône-Alpes area, in France. Your aims were to explore the added value of the “comments of the microbiologists for interpreting antimicrobial susceptibility testing (AST) profile of urinalysis that have been suggested to be suitable for guiding and improving on the quality of antibiotic prescribing by a superiority, double-blind, digital, randomized case-vignette trial. You concluded that this approach doesn’t work.

Dear Colleagues in your abstract don’t report that in your study are followed GPs mostly from the Auvergne-Rhône-Alpes area, in France, don’t explain why it is important to include “64,7% women and aged 37 years in average” and you report “multivariate analysis" a statistical analysis that isn't present the articles

Dear Collegues, in your results Lines 85-87 GPs mean age was 37 years, and the majority (64.7%) were women. Most (73.8%) of GPs were either installed or replacing GPs with an average professional experience of 11 years. Characteristics of GPs enrolled in the trial were similar between arms and are described in Table 1. All Lines in tables 1 don’t displayed the Total value and resulted puzzling.

In Table 1 your results are different

Table 1

Age

Mean ± SD

37.1 ± 11.6

36.7 ± 11.2

36.9 ± 11.4

Median (Q1 - Q3)

33.0 (29.0 - 42.0)

33.0 (29.0 - 39.5)

33.0 (29.0 - 41.0)

Professional status of GPs*

Installed

258 (60.8%)

237 (60.6%)

495 (60.7%)

Replacing other GPs

53 (12.5%)

54 (13.8%)

107 (13.1%)

Resident

113 (26.7%)

100 (25.6%)

213 (26.2%)

Main mode of practice

At hospital/healthcare facility

41 (9.67%)

32 (8.18%)

73 (8.96%)

In group practices

330 (77.8%)

318 (81.3%)

648 (79.5%)

Alone

53 (12.5%)

41 (10.5%)

94 (11.5%)

Table 2  

Lines 104-105 -106     caption is incorrect.

Lines 108, 109, 110 are different between results showed in table.

Line 108, 109, 110 are difficult to understand in table 2.

Control (N = 424)

Intervention (N = 391)

Total (N = 815)

Attitude toward AST* interpretation

At ease to interpret, willing to be trained

135 (31.8%)

112 (28.7%)

247 (30.3%)

At ease to interpret, not willing to be trained

274 (64.6%)

269 (68.9%)

543 (66.6%)

Not at ease to interpret, willing to be trained

12 (2.8 %)

9 (2.3%)

21 (2.7%)

Not at ease to interpret, not willing to be trained

3 (0.8%)

1 (0.1%)

4 (0.4%)

Dear Colleagues in Figure 1.  (Line 126) you report that “IUC-Bact: indwelling urinary catheter associated bacteriuria” but you don’t report or discuss how is possible that exist the Control and the Intervention in this very particular case.

Dear Colleagues, your caption in Figure 1 (Line 133)    is incorrect considering that in the caption presents a conclusion. “Appropriateness of antibiotic regimen, by trial arm and type of urinary tract infection”

 Dear Colleagues, it is essential that Figure 1 It is essential that Figure 1 is done again considering all the 734 GPs and not just 500 GPs, in fact in the paragrapher relate to the Statistical analysis (Lines 276, 277, 278) you assume that 500 answers would allow to reach 90% power to detect an increase in overall appropriateness. I would like to remember you that this procedure is uncorrected.

In conclusions Line 294-299 You concluded that this approach doesn’t work.

English performance

The reviewer checked the first page assuming that usually the Abstract and the first part of Introduction are carefully checked by you:

Line 16 at exploring.     Change in              to explore.

Line 18-19   --- Please change the paragraph because it is not clear. “One case-vignette with (intervention) or without (control) a ‘comment’ after AST was randomly assigned to 815 participating GPs,  

Line 20 were using please change in               used.

Line 27   did not improve the overall level of appropriateness of targeted antibiotic.

Line 27. Change UTI in UTIs in all articles, please

Line 35.       such as medicine management     correct in     such as medicinal management.

Line 40        per days                              correct in day.

English performance

The reviewer checked the first page assuming that usually the Abstract and the first part of Introduction are carefully checked by you:

Line 16 at exploring.     Change in              to explore.

Line 18-19   --- Please change the paragraph because it is not clear. “One case-vignette with (intervention) or without (control) a ‘comment’ after AST was randomly assigned to 815 participating GPs,  

Line 20 were using please change in               used.

Line 27   did not improve the overall level of appropriateness of targeted antibiotic.

Line 27. Change UTI in UTIs in all articles, please

Line 35.       such as medicine management     correct in     such as medicinal management.

Line 40        per days                              correct in day.

Reviewer 3 Report

Line 13:  originate most.... "are responsible for most of the antibiotics prescribed".  Line 17:  superiority,  use "superior".  Line 19: and aged, "64.7% women at an average age of 37 years".  Line 20: antibiotic prescribing, "prescribing antibiotics" sounds better; check other similar sentences. Also line 20, (90,1%), "should be (90.1%)". 

There is much use of the word "appropriateness" throughout the manuscript. Whereas, it is not wrong, there are other synonyms for "appropriateness"   which may fit better in a scientific publication, e.g. "suitable", "valid", "relevant", this is found throughout the manuscript. Line 38: yielding to, ""resulting in" . Also Line 38: Do you mean one "sector", or should it be "sectors".  Another word for consumption is "utilization".  Line 40: Do you mean 1000 "inhabitants", or do you mean 1000 "patients"?  Line 55&57 "inappropriate" rather than inappropriateness. Line 56: adequation?, you may mean "adherence".  Do you mean "was not effective, or suitable fro 14% of patients". Line 59: estimating,I believe you mean "to estimate" .  Line 86: "Newly appointed" rather than installed. Line 37: characteristics, do you mean "the knowledge and expertise" of GPs. Line 98 : "prescribing antibiotics", rather than prescribing antibiotics.   Table 1 is well organized, and I do not see any problems with it.  Table 2 is also well constructed and clear. Line 115:  "were ", instead of was. Also, no need for "therefore". Line 116:  343 in "the"intervention arm, and 373 in "the"......  Line 22,23 : "increase", rather than increased.   Figure 1 is clear .  Line 126 :" levels", not level, in both places.  Line 142 : should be  "case-vignette".  Table 3 is also well constructed.  Line 165 : delete "been".  Line 175 : out "to" every....  . Line 186 : use "an" AST.....  .  Line 187:  the trial and "resulted" rather than yield to.  Line 188:  focus on "the" selected dose....  .  Line 189 : There was "however", rather than thereby. Line 200 : clarify "concealment" .  Line 229 :  "complications" rather than complication. Line 231 : To insure that "the" randomization.   Figure 2 is also well constructed.  Line 265 : To "previously established" rather than last....  .Line 294 : comments by microbiologists "regarding"  rather than for......  .

This manuscript is a well documented compilation of comments by microbiologists, regarding interpretation of antimicrobial susceptibility testing, with the idea of improving therapy  in various communities. The goal is to improve management and treatment of urinary tract infections. There was some difficulty in terms of language. Just to mention, there are several synonyms for appropriateness, e.g relevance, validity, suitability, others. Though appropriateness is not wrong. The Tables and Figures are well constructed and clear. I believe it will add to significant knowledge about constructing and organizing such studies regarding infections.  

  I noted some difficulty in the English language construction, but I hope that my comments will be helpful to remedy that. Perhaps, they can also find someone who can help them with that. For example, too much use of the word "Appropriateness", which may not be wrong, but yet does not seem very scientific (at least to me). Mostly I think of "appropriate behavior", or "appropriate clothing", etc).  There are at least 7 or 8 synonyms for appropriateness, examples, relevance, validity, applicability, suitability. The other corrections I made should be a good example for rewriting the manuscript
